# Testosterone Serum Levels Are Related to Sperm DNA Fragmentation Index Reduction after FSH Administration in Males with Idiopathic Infertility

**DOI:** 10.3390/biomedicines10102599

**Published:** 2022-10-17

**Authors:** Monica Lispi, Panagiotis Drakopoulos, Giorgia Spaggiari, Francesca Caprio, Nicola Colacurci, Manuela Simoni, Daniele Santi

**Affiliations:** 1Clinical and Experimental Medicine PhD Program, University of Modena and Reggio Emilia, 41121 Modena, Italy; 2IVF Athens Centre, 10676 Athens, Greece; 3Unit of Endocrinology, Department of Medical Specialties, Azienda Ospedaliero-Universitaria di Modena, Ospedale Civile di Baggiovara, 41125 Modena, Italy; 4Department of Woman, Child and General and Specialized Surgery, University of Campania “Luigi Vanvitelli”, 80138 Naples, Italy; 5Department of Biomedical, Metabolic and Neural Sciences, University of Modena and Reggio Emilia, 42121 Modena, Italy

**Keywords:** FSH, sperm DNA fragmentation index, idiopathic male infertility, testosterone

## Abstract

Purpose: Although a robust physiological rationale supports follicle stimulating hormone (FSH) use in male idiopathic infertility, useful biomarkers to evaluate its efficacy are not available. Thus, the primary aim of the study was to evaluate if testosterone serum levels are related to sperm DNA fragmentation (sDF) index change after FSH administration. The secondary aim was to confirm sDF index validity as a biomarker of FSH administration effectiveness in male idiopathic infertility. Methods: A retrospective, post-hoc re-analysis was performed on prospectively collected raw data of clinical trials in which idiopathic infertile men were treated with FSH and both testosterone serum levels and sDF were reported. Results: Three trials were included, accounting for 251 patients. The comprehensive analysis confirmed FSH’s beneficial effect on spermatogenesis detected in each trial. Indeed, an overall significant sDF decrease (*p* < 0.001) of 20.2% of baseline value was detected. Although sDF resulted to be unrelated to testosterone serum levels at baseline, a significant correlation was highlighted after three months of FSH treatment (*p* = 0.002). Moreover, testosterone serum levels and patients’ age significantly correlated with sDF (*p* = 0.006). Dividing the cohort into responders/not responders to FSH treatment according to sDF change, the FSH effectiveness in terms of sDF improvement was related to testosterone and male age (*p* = 0.003). Conclusion: Exogenous FSH administration in male idiopathic infertility is efficient in reducing sDF basal levels by about 20%. In terms of sDF reduction, 59.2% of the patients treated were FSH-responders. After three months of FSH administration, a significant inverse correlation between sDF and testosterone was detected, suggesting an association between the FSH-administration-related sDF improvement and testosterone serum levels increase. These observations lead to the hypothesis that FSH may promote communications or interactions between Sertoli cells and Leydig cells.

## 1. Introduction

### 1.1. Idiopathic Male Infertility

The contributing role of the male partner in infertile couple is currently well demonstrated and estimated to be causative in about 50% of cases [1]. Recently, the International Committee for Monitoring Assisted Reproductive Technologies (ICMART) provided a new definition of hypogonadotropic hypogonadism that fits well with the male infertility status, namely, “Gonadal failure associated with reduced gametogenesis and reduced gonadal steroid production due to reduced gonadotropin production or action” [2]. Male idiopathic infertility is defined as the clinical condition in which spermatogenesis is altered without a known cause [3,4,5]. This condition should be considered distinct from unexplained male infertility, defined as infertility of unknown origin but with normal sperm parameters [5]. Male idiopathic infertility management remains complex and challenging since an aetiological therapy cannot be proposed. Several empirical treatments have been suggested, trying to mirror the therapeutic approach applied to hypogonadotropic hypogonadism [6,7,8]. Among these, the exogenous administration of follicle stimulating hormone (FSH) shows a strong physiological rationale when steroidogenesis is not compromised, aiming at stimulating spermatogenesis [1,7,9]. Although FSH administration is suggested by several scientific societies’ guidelines, this hormonal approach is currently used only in few countries worldwide due to national regulations about drug reimbursement [1,7,9,10]. 

### 1.2. FSH Administration in Male Idiopathic Infertility

In male idiopathic infertility, FSH administration could be considered either as a replacement therapy or as an attempt to boost spermatogenesis. Indeed, FSH-mediated testicular overstimulation activates spermatogenesis and, theoretically, may improve sperm quality, as reported in 21 clinical trials and four meta-analyses [11,12,13,14]. The results of these meta-analyses are obviously affected by the high heterogeneity of included trials and the selection of pregnancy rate as primary endpoint. Although pregnancy rate is clearly the most meaningful outcome when evaluating an infertile couple, it may be affected by both male and female factors, irrespective of FSH action on the male partner. Therefore, FSH action on male cannot be assessed without considering the different confounders. An overall pregnancy rate increase (OR: 2.09; CI95%: 1.46, 3.01) in infertile couples in which FSH was administered to the male partner was detected. However, the calculated number needed to treat was high, showing that 10 to 18 men should be treated to achieve one pregnancy, spontaneously and following the assisted reproductive technique (ART), respectively [13,15]. In addition, only a limited increase in conventional sperm parameters was detected after FSH administration [13,15], confirming the debate on low accuracy of conventional semen analysis to estimate hormonal treatment efficacy. Thus, the first crucial question about FSH efficacy is to evaluate which could be the most appropriate endpoint.

### 1.3. Endpoints to Assess Treatment Efficacy Are Not Satisfactory

Alongside sperm concentration improvement, several randomized controlled clinical trials (RCTs) showed sperm DNA damage reduction in men treated with FSH [16]. This sperm quality parameter was evaluated by the sperm DNA fragmentation (sDF) index, which is a simple and reliable tool, currently suggested as a co-endpoint in male infertility work-up [17,18]. Infertile men showed a higher sDF index compared with fertile ones, and FSH administration reduces sDF by about 4.24% in idiopathic infertility [16,17]. This relative sDF improvement was higher than the sperm concentration increase reported in the aforementioned meta-analysis (2.66 million/mL; CI95%: 0.47, 4.84). However, it is still not clear whether sDF improves after FSH administration only in men with altered baseline levels. Similarly, we need to understand if sDF could be considered as a marker of FSH efficacy, discriminating between patient “responders” and “non-responders” to FSH treatment.

It is well established that in order to identify markers to evaluate FSH efficacy in male idiopathic infertility, RCTs remain the best methodological approach. However, since RCTs are very difficult in this field due to the large amount of resources, time and expertise required, it has been suggested that the re-analysis of raw RCTs-derived data could give an answer to new study questions [19]. 

### 1.4. Study Objectives

Since FSH acts on Sertoli cells, researchers focused their attention to the seminiferous testicular compartment, without considering the interstitial counterpart. However, aiming to identify new markers of FSH effectiveness, the potential correlation between semen parameters and steroids after FSH stimulation is still an unexplored aspect with potential relevant clinical impact. Thus, the aim of our study was the evaluation of the relationship between sDF and testosterone serum levels in men treated with FSH. We wished to explore whether a communication/interaction between the two cell compartments of the testes (Sertoli and Leydig cells) could be found by exogenous FSH administration.

Moreover, secondary aims were (i) to confirm FSH efficacy in terms of sDF improvement and (ii) to evaluate whether testosterone serum levels could predict sDF change after FSH administration.

## 2. Materials and Methods

### 2.1. Systematic Literature Search

A retrospective *post*-*hoc* analysis using Individual Patients’ Data (IPD) from previously published clinical trials was conducted. Thus, the study consisted of two steps: in the first one, a systematic search of the literature was performed; in the second step, data derived from the clinical trials identified were analysed.

During the first step, a comprehensive systematic literature search for English-language articles in MEDLINE (PubMed) and EMBASE was conducted. The literature search was performed using the following keywords: male infertility, couple infertility, FSH, FSH administration, sperm DNA fragmentation index, sDF and testosterone. The Boolean functions AND and OR were used to combine keywords.

The following inclusion criteria were established before the literature search: (i) clinical trials (ii) in which the male partner of infertile couples was treated with FSH and both (iii) sDF index and (iv) testosterone serum levels were reported. Considering the rich literature on FSH application in male idiopathic infertility, keywords and inclusion criteria were set to detect only those studies in which both testosterone and sDF after treatment were reported.

Men enrolled showed idiopathic infertility, with one or more sperm parameters altered, in whom no specific causes of male infertility were detected. Retrospective studies were not included. No other inclusion/exclusion criteria were provided. 

The corresponding author of each eligible trial was contacted to obtain raw data. When the author accepted, IPD were collected, considering the following specific endpoints: sDF index; testosterone serum levels (measured both before and after FSH administration); patient’s age; body mass index (BMI); FSH dosages; treatment duration; conventional semen analysis parameters (such as sperm concentration, total sperm count, progressive sperm motility, total sperm motility and sperm morphology); hormonal evaluations (such as FSH, luteinizing hormone (LH), inhibin B, sex hormone binding globulin (SHBG) and anti-Mullerian hormone (AMH)); couple infertility duration; number of pregnancies obtained, both spontaneous and after assisted reproduction (if available). 

IPD obtained by each study were combined in a single dataset and descriptive analyses were conducted. 

### 2.2. Identification of FSH Administration Efficacy in Men with Idiopathic Infertility

The final dataset generated on IPD extracted following the systematic literature search was evaluated to reach primary and secondary endpoints. In detail, the primary endpoint was the potential correlation between testosterone serum levels and sDF change after FSH treatment. Secondary endpoints were (i) sDF decrease after FSH administration and (ii) to determine whether testosterone serum levels could predict the sDF decrease after FSH administration.

#### 2.2.1. Endpoints’ Definitions

As previously reported, the efficacy of FSH administration is associated with sDF index decrease. This improvement was recorded in each study considered alone (or “as a unit”). Here, we analysed IPD to quantify the comprehensive sDF decrease after FSH administration, comparing pre- and post-treatment values.

Similarly, the pre- and post-FSH treatment change in secondary endpoints was evaluated. 

In order to better define a successful treatment, the entire cohort was divided into responders and non-responders to FSH. Responders were empirically defined as men in whom the sDF index decreased by at least 20% (relative decrease) of baseline levels after treatment. This threshold was empirically adopted to highlight a clinically significant sDF reduction.

#### 2.2.2. Statistical Analysis

Statistical procedures were applied to the dataset to reach the secondary objectives of the study.

In order to determine the correlation between testosterone serum levels and sDF change after FSH treatment, correlation analyses were performed, combining testosterone serum levels and sDF at baseline and after FSH administration. Data were first analysed for distribution with Kolmogorov–Smirnov test and correlations were assessed using Pearson’s or Spearman’s methods for normally or not-normally distributed data, respectively. Correlation analyses were performed by considering anthropometric variables, hormones, semen parameters and sDF index, and by applying Bonferroni adjustment. Since 14 variables were considered, *p* < 0.003 was considered for statistical significance in correlation analyses. Moreover, multivariate stepwise linear regression analyses were performed, using sDF index as the dependent variable and testosterone, FSH, LH, SHBG, inhibin B, AMH, FSH treatment duration, patient’s age and BMI as independent parameters. In order to correct potential confounders, multiple models were used to yield total-effect estimates for covariates [20].

Moreover, to determine the change in hormone levels to classify a man with and without significant sDF decrease, the entire cohort was considered by dividing patients into responders and non-responders, according to sDF change after FSH administration. Then, logistic regression analyses were performed using the responders/non-responders classification as the dependent variable. Conventional semen parameters, patients’ age, BMI, FSH dosage, treatment duration and hormones collected after FSH administration were included among co-variates. Logistic regression analyses were repeated considering baseline parameters to identify potential predictors of FSH effectiveness. In this setting, to graphically show the connection/trade-off between clinical sensitivity and specificity for every possible cut-off for a predictor of FSH effectiveness, receiver operating characteristic (ROC) curves were generated.

Statistical analysis was performed using the “Statistical Package for the Social Sciences” software for Windows (version 27.0; SPSS Inc., Chicago, IL, USA). For all comparisons, *p* < 0.05 was considered statistically significant.

## 3. Results

Among 21 published studies investigating FSH administration to the male partner of infertile couples, three studies were selected (Appendix A) according to the inclusion criteria specified above [21,22,23]. Table 1 summarizes the study characteristics.

Finally, 251 patients were overall considered. Table 2 summarizes baseline patients’ characteristics.

### 3.1. Are Testosterone Serum Levels Correlated to sDF Decrease after FSH Administration?

At baseline, sDF was not significantly related to testosterone serum levels, which, in turn, were only directly related to LH serum levels (Spearman’s correlation analysis: Rho 0.362, *p* < 0.001) (Appendix A). FSH serum levels were directly related to both LH (*p* < 0.001) and inhibin B serum levels (*p* < 0.001) (Appendix A).

After FSH treatment, sDF was inversely and significantly correlated to testosterone serum levels (Rho −0.327, *p* = 0.002) (Appendix A). Moreover, testosterone serum levels remained statistically significantly correlated with LH (Rho 0.272, *p* = 0.004) (Appendix A).

In order to identify how study variables correlated with both sDF and testosterone serum levels after FSH administration, multivariate stepwise linear regression analysis was performed. The sDF index obtained after three months of FSH administration was directly related to testosterone serum levels and inversely to patients’ age (*p* = 0.006) (Table 3).

### 3.2. What Is the Overall sDF Decrease after FSH Administration?

The comprehensive analysis confirmed the FSH beneficial effect on spermatogenesis detected in each trial. Indeed, increasing the sample size, this new analysis highlighted an overall significant sDF decrease (*p* < 0.001) of 20.2% of the baseline value (Table 4). Moreover, FSH administration significantly increased inhibin B (*p* = 0.006), AMH (*p* = 0.001) and testosterone (*p* = 0.001) serum levels (Table 4).

### 3.3. Are Testosterone Serum Levels Predictive of Responders and Non-Responders, Defined on sDF Decrease after FSH Administration?

Here, we empirically defined FSH-responders as men who achieved an sDF reduction of at least 20% of baseline values after three months of FSH administration. With this definition, the effectiveness rate was 59.2% (148 patients out of 251). Responders showed higher testosterone serum levels after FSH treatment (*p* = 0.014) (Table 5); however, conventional semen parameters were not significantly different between responders and non-responders (Table 5).

Logistic regression analyses were performed to highlight markers and predictors of response to FSH treatment. The first analysis used all parameters detected after FSH administration as cofactors/covariates, showing a significant relationship between response and both testosterone and male age serum levels (Table 6). To identify potential thresholds of these variables, two ROC analyses were generated using testosterone and age as test variables. Neither testosterone (area under the curve—AUC = 0.584, *p* = 0.124) nor age (AUC = 0.537, *p* = 0.373) displayed significant thresholds.

The second analysis considered all parameters detected at baseline as cofactors/covariates. A significant predictive role of FSH efficacy was detected only for sDF (*p* = 0.002) (Table 7). An ROC analysis was performed, setting the response as the dependent variable and sDF as the test variable. The ROC generated showed a significant threshold (AUC = 0.754, *p* < 0.001) of 16.75% (sensitivity 75.7%, specificity 69.1%) (Figure 1). When patients with baseline sDF higher than 16.75% were selected, 72.7% (104 patients) showed an sDF decrease of at least 20% of baseline sDF levels, with an average decrease of 5.23% (from 24.1 ± 7.2% to 18.9 ± 5.9%, *p* < 0.001). Interestingly, in this subgroup, a significant total testosterone serum levels increase was detected (from 4.2 ± 2.3 to 4.9 ± 2.0 ng/mL, *p* = 0.003). Similarly, a significant inhibin B and AMH increase after FSH administration was confirmed (*p* = 0.034 and *p* = 0.001, respectively).

## 4. Discussion

This re-analysis of published clinical trials data investigating FSH administration in male idiopathic infertility highlights a novel aspect of male infertility management. Alongside the expected amelioration of conventional semen quality and an overall 20% decrease of sDF baseline index, exogenous FSH stimulation induces an increase in testosterone, inhibin B and AMH serum levels. Both inhibin B and AMH are Sertoli cell products, reflecting the proliferative status of the testicular germinative epithelium [25,26]. Thus, their increase after FSH administration is not unexpected. On the contrary, the testosterone rise after FSH administration is unexpected and detected here for the first time, suggesting an action of FSH on testicular function more complex than thought so far. Indeed, we could speculate that FSH boosts spermatogenesis throughout a direct action on Sertoli cells and also an indirect effect involving the testis interstitial compartment. This finding is in line with previous demonstrations of the capability of the supraphysiological FSH stimulation to sustain spermatogenesis even in the absence of LH action [27]. Indeed, a quali-quantitative normal spermatogenesis was reported in a hypophysectomised man presenting an activating FSH receptor mutation, suggesting that a strong FSH action alone could support the LH/testosterone function [27].

Our new analysis showed for the first time the global testicular action (not limited to the spermatogenic compartment) of FSH chronically administered to infertile men. As confirmed, after three months of FSH administration, sperm quality (in terms of sDF index) correlated with testosterone serum levels, highlighting the association between the FSH-related sDF improvement and the increase in testosterone serum levels. Accordingly, both multivariate linear and logistic regression analyses identified a strong correlation between testosterone increase and sDF decrease after FSH treatment. This finding is totally novel since the literature on this topic is entirely silent. This innovative result could open new perspectives in the way of evaluating responses to FSH treatment in male idiopathic infertility. This new correlation, although interesting, is far from being directly transposed to clinical practice. Specific, properly designed prospective trials must be designed to understand the real clinical application of the finding. From a physiological point of view, the strict connection between seminiferous and interstitial testicular compounds is expected, since effective spermatogenesis requires both FSH action and adequate intratesticular testosterone levels [28]. However, this link has been generally underestimated in clinical practice, since intratesticular testosterone assessment is very complex, requiring testicular biopsy or sampling [29]. In addition, testosterone measured in the peripheral blood correlates poorly to its intratesticular levels, which are estimated to be at least 100 times higher [30]. For these reasons, several attempts have been made to identify surrogate markers of intratesticular testosterone levels [29]. Among these, serum 17-hydroxyprogesterone (17-OHP) and Insulin-like factor 3 (INSL3) have been proposed [30,31]. These hormones were demonstrated as able to predict intratesticular testosterone levels after human chorionic gonadotropin (hCG) stimulation [32,33]. However, both 17-OHP and INSL3 did not correlate with testosterone intratesticular levels at baseline, only after hCG treatment. This result mirrors the sDF–testosterone correlation detected only after FSH administration in this study. Thus, the connection of sDF–testosterone is novel in the field of human reproduction, although some suggestion of such a correlation is provided in other fields. For example, Wood et al. described improvements in both testosterone serum levels and sDF index six months after bariatric surgery [34]. In our analysis, we speculate that both testicular compartments, seminiferous and interstitial, tend to realign only after overstimulation induced by exogenous gonadotropins. However, as neither intratesticular testosterone levels nor their surrogate markers are available in our analysis, we cannot provide conclusive explanations of the FSH action at intratesticular level.

Putting together three different cohorts of patients treated with the same regimen of recombinant FSH for three months, we highlight an overall 20% decrease of sDF baseline index with the current therapeutic approach. In particular, we detect a FSH efficacy rate of about 59.2%, considering the sDF index decrease. This relevant result could be directly translated into clinical practice. Hitherto, many direct tests have been suggested to predict sperm capability to penetrate the oocyte, such as sperm–zona binding ratios and zona pellucida-induced acrosome reaction tests [35,36]. Similarly, indirect variables could be measured in seminal plasma with the same objective, such as phospholipase, sperm acrosin, fructose and neural alfa-glucosidase [37,38,39]. In this setting, the sDF index provides an informative and reliable measure of the real fertilization capability [40,41,42,43]. SDF is the end result of the action of multiple factors that induce single- or double-strand DNA breaks in the sperm genome [44], due to oxidative stress [45], apoptosis [46,47], impaired chromatin remodelling [48] and environmental agents, such as toxicants, drugs and radiation [49,50,51]. As a result, high levels of sDF index reflect an impaired semen quality [17] and a reduced fertility [16]. Sperm DNA integrity is crucial for embryo development and successful pregnancy outcome, and sDF values are inversely related to the chances of achieving natural pregnancy [52,53]. Accordingly, increased sDF index was comprehensively detected in infertile compared with fertile men in a recent meta-analysis, which proposed an sDF threshold of 20% to discriminate between fertile and infertile men (AUC: 0.84, *p* < 0.001, sensitivity 79%, specificity 86%) [16]. The proven relevance of sDF assessment in human fertility justified the insertion of its measurement within the latest edition of the World Health Organization (WHO) laboratory manual for the examination and processing of human semen [54]. In the current re-analysis, sDF index significantly decreased after FSH treatment concomitantly with the increase in conventional semen parameters, demonstrating an overall FSH-induced improvement in sperm quality, which could be measured through this relatively new tool. However, the correlation between sDF and conventional semen parameters is still controversial [55,56,57]. Here, sDF is correlated to sperm progressive motility after FSH administration, in line with previous studies depicting the highest sDF index in men presenting the lowest percentage of motile sperms [58,59]. A connection between sDF and sperm motility could be hypothesized since these two parameters share a marked detrimental susceptibility to reactive oxygen species (ROS). Indeed, oxidative stress is assumed to be the most relevant causative factor contributing to sDF [17], while sperm motility is acquired during the long sperm transit through the epididymis, resulting more likely vulnerable to ROS damage [47].

Subgrouping our cohort in responders and non-responders to FSH treatment considering an sDF decrease threshold of 20% of its basal level could be useful to identify markers and predictors of FSH efficacy. With this approach, only two markers of FSH efficacy are identified, testosterone serum levels and male age, although ROC analyses were not able to identify useful thresholds for clinical practice. On the contrary, sDF amelioration after FSH therapy is predicted only by sDF basal levels, with a significant threshold of 16.75%. Since no differences between responders and non-responders were demonstrated by conventional semen parameters, we could speculate that semen analysis is not really accurate to evaluate male fertility status.

Our results should be carefully considered due to several limitations. First, we combined raw data of three clinical studies, putting together a large cohort of patients for this research topic, yet still limited to elaborate definitive conclusions. Second, a relatively high heterogeneity among trials should be expected. Although all three trials used recombinant FSH at the same dosage (150 IU every other day) for the same treatment period (three months), inclusion criteria of each trial may have differed. Indeed, Simoni et al. enrolled only men with sDF >15% at baseline, while Colacurci et al. did not consider sDF index as an inclusion criterion. Thus, enrolled populations were not completely homogeneous. In addition, in all studies, testosterone serum levels were measured by immunometric assays and not by liquid chromatography-mass spectrometry, which is the gold standard for steroid measurement [60].

## 5. Conclusions

In conclusion, our analysis reports for the first time an association between testosterone levels rise and sDF improvement, after exogenous FSH administration in men with idiopathic infertility. Both testosterone and male age levels represent FSH-effectiveness markers, although clinically applicable thresholds were not identified. These results deserve further consideration, but broaden the vision on male idiopathic infertility management, suggesting that fertility status and treatments response should not be barely limited to conventional semen parameters assessment. On the other hand, sDF baseline levels are predictors of FSH response, and those men presenting basal sDF index > 16.75% are expected to be more sensitive to FSH-induced sDF amelioration. How this improvement could translate into clinically relevant outcomes, i.e., pregnancy rates, remains to be investigated.

## Figures and Tables

**Figure 1 biomedicines-10-02599-f001:**
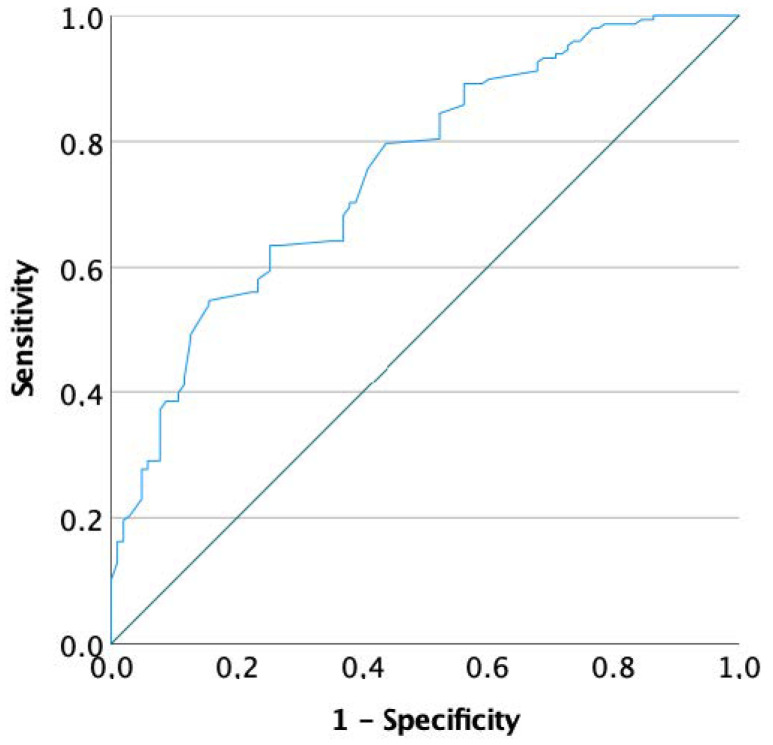
Receiver operating curve (ROC) analysis using responders as dependent variable and sperm DNA fragmentation (sDF) index as test variable.

**Table 1 biomedicines-10-02599-t001:** Characteristics of clinical trials included in the analysis in which male partners of infertile couples were treated with follicle stimulating hormone (FSH).

Author	Year	Study Design	FSH Type	FSH Scheme	Inclusion Criteria	Number of Patients (n)		Age (Years) (Mean ± SD)	
						Study Group	Control Group	Study Group	Control Group
Colacurci et al.	2012	Prospective longitudinal, randomized, case-control	r-FSH	150 IU on alternate days for 90 days	FSH 1–7 IU/L, LH 1–8 IU/L, T 3–10 ng/mL	65	64	31.6 ± 3.1	33.6 ± 3.5
Simoni et al.	2016	Prospective longitudinal, case-control	r-FSH	150 IU on alternate days for 90 days	FSH ≤8 IU/l, normal LH and T, homozygous FSHR p.N680S N or S genotype, sDF index >15%	66	-	36.4 ± 4.7	-
Colacurci et al.	2018	Prospective longitudinal, case-control	r-FSH	150 IU on alternate days for 90 days	FSH 1–8 IU/L, sperm count >10 million, total sperm motility 5–25%	111	-	36.1 ± 4.7	-

FSHR: follicle stimulating hormone receptor; N: asparagine; r-FSH: recombinant follicle stimulating hormone; S: serine; SD: standard deviation; sDF: sperm DNA fragmentation; T: testosterone.

**Table 2 biomedicines-10-02599-t002:** Baseline patient’s characteristics. Data are expressed as median (interquartile range).

Variables	Reference Ranges *	Baseline Values
Age (years)	-	35.0 (6.0)
BMI (kg/m^2^)	-	25.5 (3.5)
FSH (IU/L)	1–8	3.1 (1.9)
LH (IU/L)	1–8	2.9 (1.4)
Prolactin (ng/mL)	3–13	7.9 (4.0)
Testosterone (ng/mL)	>3.0	4.0 (1.7)
Semen volume (mL)	>1.5	2.2 (1.8)
Sperm concentration (million/mL)	>15	24.0 (66.5)
Total sperm number (million)	>39	64.3 (237.1)
Progressive motility (%)	>32	20.8 (9.0)
Normal morphology (%)	>4	14.0 (9.0)
Sperm DNA fragmentation index (%)	-	18.0 (12.5)
Sex hormone binding globulin (nmol/L)	-	32.0 (16.1)
Inhibin B (pg/mL)	-	137.3 (70.0)
Anti-Mullerian hormone (ng/mL)	-	3.7 (3.4)
Smokers *n* (%)	-	120 (47.8)

BMI: body mass index; FSH: follicle stimulating hormone; LH: luteinizing hormone. * Reference ranges for semen analysis were evaluated considering the V edition of the World Health Organization manual for semen analysis [24].

**Table 3 biomedicines-10-02599-t003:** Multivariate stepwise linear regression analyses using sperm DNA fragmentation index after three months of follicle stimulating hormone (FSH) treatment as dependent variable. Bold characters reported statistically significant parameters.

	Not Standardized Coefficients		Standardized Coefficient	t	*p*-Value	95% Confidence Interval	
	Beta	Standard Error	Beta			Lower Limit	Upper Limit
(Constant)	9.988	10.565	-	0.945	0.348	−11.112	31.088
Testosterone	−3.277	1.722	−0.258	−2.054	**0.003**	−6.781	−0.931
Age	4.291	1.155	0.220	1.931	**0.004**	0.410	2.311
BMI	0.188	0.183	0.078	1.025	0.307	−0.174	0.549
FSH	−0.004	0.423	−0.001	−0.011	0.992	−0.839	0.830
LH	0.438	0.493	0.074	0.888	0.375	−0.535	1.411
SHBG	0.031	0.062	0.049	0.502	0.617	−0.091	0.154
Inhibin B	−0.001	0.009	−0.008	−0.092	0.927	−0.018	0.016
AMH	0.214	0.169	0.096	1.269	0.206	−0.119	0.547
FSH duration	0.293	0.215	0.146	1.365	0.177	−0.136	0.722

AMH: anti-Mullerian hormone; BMI: body mass index; FSH: follicle stimulating hormone; LH: luteinizing hormone; SHBG: sex hormone binding globulin.

**Table 4 biomedicines-10-02599-t004:** Comparison between baseline and after follicle stimulating hormone (FSH) administration, considering both hormone and semen parameters, using Mann–Whitney *U*-test. Data are expressed as mean ± standard deviation. Bold characters reported statistically significant parameters.

Variables	Baseline	After FSH Administration	*p*-Value
sDF index (%)	18.9 ± 8.6	15.8 ± 6.5	**<0.001**
Testosterone (ng/mL)	4.3 ± 2.2	4.9 ± 1.8	**0.001**
FSH (IU/L)	3.3 ± 1.5	5.6 ± 2.1	**<0.001**
LH (IU/L)	3.1 ± 1.4	2.9 ± 1.3	0.098
Prolactin (ng/mL)	8.2 ± 3.6	8.3 ± 3.8	0.783
SHBG (nmol/L)	32.0 ± 12.6	32.5 ± 12.3	0.629
Inhibin B (pg/mL)	154.5 ± 69.6	175.4 ± 79.9	**0.006**
Anti-Mullerian hormone (ng/mL)	4.3 ± 2.8	5.4 ± 4.0	**0.001**
Semen volume (mL)	2.8 ± 2.5	3.1 ± 1.3	0.058
Sperm concentration (million/mL)	59.8 ± 63.3	95.4 ± 107.2	**<0.001**
Total sperm number (millions)	217.7 ± 522.8	323.3 ± 420.2	**0.019**
Progressive motility (%)	20.3 ± 7.5	39.8 ± 16.2	**<0.001**
Normal morphology (%)	16.0 ± 11.5	21.3 ± 13.3	**<0.001**

FSH: follicle stimulating hormone; LH: luteinizing hormone; sDF: sperm DNA fragmentation; SHBG: sex hormone binding globulin.

**Table 5 biomedicines-10-02599-t005:** Comparison between responders and non-responders based on sperm DNA fragmentation index, considering parameters after three months of follicle stimulating hormone (FSH) administration using Mann–Whitney *U*-test. Data are expressed as mean ± standard deviation. Bold characters reported statistically significant parameters.

	Non-Responders	Responders	*p*-Value
Number of patients n (%)	102 (40.8)	148 (59.2)	**-**
Age (years)	36.1 ± 4.5	34.9 ± 5.3	0.070
sDF index (%)	17.2 ± 6.1	15.5 ± 7.0	**0.007**
Testosterone (ng/mL)	4.5 ± 1.7	5.1 ± 1.9	**0.014**
FSH (IU/L)	5.7 ± 2.3	5.5 ± 2.0	0.362
LH (IU/L)	3.1 ± 1.4	2.7 ± 1.2	0.125
Prolactin (ng/mL)	9.5 ± 4.6	7.6 ± 3.1	0.114
SHBG (nmol/L)	31.5 ± 12.7	33.0 ± 11.9	0.389
Inhibin B (pg/mL)	162.6 ± 74.5	182.8 ± 80.8	0.121
AMH (ng/mL)	5.1 ± 3.6	5.6 ± 4.3	0.404
Semen volume (mL)	2.9 ± 1.2	3.2 ± 1.3	0.129
Sperm concentration (million/mL)	81.6 ± 96.2	104.0 ± 112.8	0.140
Total sperm number (millions)	263.2 ± 381.0	360.7 ± 439.9	0.100
Progressive motility (%)	40.8 ± 15.5	39.7 ± 16.6	0.440
Normal morphology (%)	19.5 ± 13.7	22.2 ± 13.1	0.141

AMH: anti-Mullerian hormone; FSH: follicle stimulating hormone; LH: luteinizing hormone; sDF: sperm DNA fragmentation index; SHBG: sex hormone binding globulin.

**Table 6 biomedicines-10-02599-t006:** Logistic regression analysis to predict patients with sperm DNA fragmentation index improvement. Bold characters reported statistically significant parameters.

	B	Standard Error	Wald	*p*-Value	Exp(B)	95% Confidence Interval	
						Lower Limit	Upper Limit
Intercept	−0.298	4.671	1	0.949	-	-	-
sDF	0.027	0.054	0.247	0.619	1.027	0.924	1.143
Age	3.387	0.385	2.163	**0.005**	1.066	1.006	1.399
FSH	−0.11	0.196	0.314	0.575	0.896	0.610	1.316
Prolactin	0.107	0.081	1.725	0.189	1.113	0.949	1.305
LH	0.387	0.289	1.793	0.181	1.472	0.836	2.592
Testosterone	4.121	0.305	3.982	**0.015**	3.822	1.670	4.521
SHBG	0.002	0.035	0.002	0.964	1.002	0.935	1.073
Inhibin B	−0.002	0.004	0.297	0.586	0.998	0.989	1.006
AMH	−0.045	0.085	0.275	0.600	0.956	0.809	1.130
Semen volume	−0.31	0.423	0.538	0.463	0.733	0.32	1.680
Sperm concentration	−0.002	0.012	0.038	0.846	0.998	0.975	1.021
Total sperm number	0.001	0.003	0.043	0.835	1.001	0.994	1.007
Sperm motility	0.028	0.023	1.575	0.210	1.029	0.984	1.075
Sperm morphology	−0.014	0.028	0.237	0.626	0.986	0.933	1.043
BMI	−0.063	0.100	0.396	0.529	0.939	0.772	1.142
Smoking	0.731	0.699	1.096	0.295	2.078	0.528	8.171

AMH: anti-Mullerian hormone; BMI: body mass index; FSH: follicle stimulating hormone; LH: luteinizing hormone; SE: standard error; SHBG: sex hormone binding globulin.

**Table 7 biomedicines-10-02599-t007:** Predictive markers: logistic regression analysis performed using sperm DNA fragmentation index improvement as dependent variable and all parameters obtained at baseline as independent ones.

	B	Standard Error	Wald	*p*-Value	Exp(B)	95% Confidence Interval	
						Lower Limit	Upper Limit
Intercept	12.098	5.698	1	0.034	-	-	-
sDF	−0.206	0.066	9.792	**0.002**	0.814	0.715	0.926
Age	−0.129	0.073	3.101	0.078	0.879	0.762	1.015
FSH	−0.447	0.320	1.95	0.163	0.64	0.341	1.198
Prolactin	0.169	0.103	2.693	0.101	1.184	0.968	1.448
LH	0.137	0.302	0.205	0.650	1.147	0.634	2.074
Testosterone	0.287	0.33	0.754	0.385	1.332	0.697	2.544
SHBG	−0.07	0.04	3.09	0.079	0.932	0.862	1.008
Inhibin B	−0.006	0.006	0.914	0.339	0.994	0.982	1.006
AMH	−0.046	0.144	0.100	0.752	0.955	0.720	1.268
Semen volume	0.113	0.389	0.084	0.772	1.119	0.523	2.397
Sperm concentration	−0.002	0.017	0.013	0.908	0.998	0.965	1.032
Total sperm number	0.001	0.005	0.032	0.858	1.001	0.992	1.010
Sperm motility	0.103	0.06	2.937	0.087	1.108	0.985	1.246
Sperm morphology	0.040	0.041	0.92	0.337	1.040	0.960	1.128
BMI	−0.255	0.126	4.111	0.043	0.775	0.605	0.992
Smoking	1.385	0.738	3.524	0.060	3.995	0.941	16.962

AMH: anti-Mullerian hormone; BMI: body mass index; FSH: follicle stimulating hormone; LH: luteinizing hormone; SE: standard error; SHBG: sex hormone binding globulin.

## Data Availability

Data could be available after request to authors.

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
