# Peer review of "Testosterone Serum Levels Are Related to Sperm DNA Fragmentation Index Reduction after FSH Administration in Males with Idiopathic Infertility"

_biomedicines, 2022, doi:10.3390/biomedicines10102599_

Round 1
Reviewer 1 Report
The patient group is idiopathic, that Means the men have normal semen parameters. How would you correlate FSH administration and sperm DNA?
This paper summarize the literature information regarding FSH administration in idiopathic male infertility.
In this manuscript, There is a hypothetical Issue on causation vs correlation. The researchers should identify biological mechanism behind this hypothesis. How does FSH administration affect the sperm DNA at tissue or cellular level. Otherwise literature analysis only can't identify the significancy of the FSH administration.
3 clinical study and no biological hypothesis doesn't seem satisfactory to make a conclusion.
Idiopathic group is infertile group with normal semen parameters. So, FSH administration causes sperm DNA fragmentation so it eventually result in infertility is not correct.
Author Response
ANSWER: The patients of the three trials included showed an idiopathic infertility. To better focus the study on this topic, we introduced the definition of idiopathic infertility, highlighting differences with unexplained infertility, as follows:
“Male idiopathic infertility is defined as the clinical condition in which spermatogenesis is altered without a known cause [3-5]. This condition should be considered separately to unexplained male infertility, defined as infertility of unknown origin but with normal sperm parameters [5]. Male idiopathic infertility management remains complex and challenging since an aetiological therapy cannot be proposed.”.
In particular, idiopathic infertility does not means that they have normal semen parameters. Indeed, looking at Table 1, inclusion criteria for each of the three studies are reported. Only in one study semen analysis is accounted among inclusion criteria, reporting thresholds below those considered as reference from the WHO manual. Indeed, all studies included patients suffering of couple infertility, without any other known causes in the male partner. The terms “idiopathic” refers to the lack of known cause of the infertility detected.
In this setting, the potential beneficial action of FSH, in terms of sDF decrease has been already suggested in the literature and also meta-analysed (Santi D, Spaggiari G, Simoni M. Reprod Biomed Online. 2018 Sep;37(3):315-326. doi: 10.1016/j.rbmo.2018.06.023).
Our work was an original work based on re-analysis of data previously collected in different clinical settings. The main result is the correlation between sDF and testosterone after FSH administration. This is just a correlation and not a causative effect. However, since this correlation was not present at baseline and since FSH administration significantly decreases sDF, we speculated that this new correlation could be the result of FSH administration.
These new findings deserve further investigations, and the present study is not designed to assess a clear-cause-effect relationship that should be addressed by exploring potential biological mode of action. In the discussion section, we indeed only speculated about the role of FSH on testosterone serum levels. We reported that “…the testosterone raise after FSH administration is unexpected and detected here for the first time, suggesting an action of FSH on testicular function more complex than thought so far. Indeed, we could speculate that FSH boosts spermatogenesis throughout a direct action on Sertoli cells and also an indirect effect involving the testis interstitial compartment. This finding is in line with previous demonstrations of the capability of the supraphysiological FSH stimulation to sustain spermatogenesis even in the absence of LH action [26]. Indeed, a quali-quantitative normal spermatogenesis was reported in a hypophysectomised man presenting an activating FSH receptor mutation, suggesting that a strong FSH action alone could support the LH/testosterone function [26].”. This is a potential hypothesis to explain this novel result that deserve, as mentioned, an extensive well designed biological investigation

Reviewer 2 Report
Reviewer’s report
Date: 22 August 2022
Testosterone serum levels are related to sperm DNA fragmentation index reduction after FSH administration in males with idiopathic infertility
Reviewer’s report:
In the manuscript titled Testosterone serum levels are related to sperm DNA fragmentation index reduction after FSH administration in males with idiopathic infertility, the authors performed a re-analysis of clinical trials data investigating the role of FSH supplementation in DNA fragmentation in men with idiopathic infertility. The manuscript may be useful for the advancement of the field; however, it needs some revisions prior to its acceptance for publication. The most relevant commentary relates to the methods section and requires a reevaluation of the study design prior to publication. Detailed comments are as follows:
· Introduction:
o Section 1.1 – the mention of hypogonadotropic hypogonadism is out of place; how can this be tied into idiopathic male infertility in this section?
o Section 1.2 – this section contains a useful overview of prior literature but is a bit too long.
o Section 1.3 – what are the clinical implications of a high or low sDF index?
o Overall, the introduction is insufficient, mostly due to structure and wording. A better way to structure the introduction would be to describe the physiologic role of FSH, and the pathophysiology of DNA fragmentation and then bring it together as a “we don’t know how these two might be related”
· Methods:
o Please mention why these keywords were chosen and why these inclusion criteria were used.
o The use of “systematic review” is confusing in this context. It is unclear if this systematic literature review was how the mechanism by which the clinical trials were selected. If so, clarify. If it is truly a systematic review, the studies included should be evaluated and an evaluation of these studies should be provided in the paper.
o How does this study differ from a meta-analysis? If it is a meta-analysis, please rewrite it, but if it is a systematic review it needs to have the appropriate metrics included.
o Breaking this section into so many subsections is difficult to read. Unless this is required by the journal, please adjust to make larger, more cohesive sections.
· Results:
o Please include a PRISMA diagram
o Please describe which analysis was used to determine the values reported in the first two paragraphs of section 3.1 (if it was all from the correlation analysis, can be mentioned once)
· Discussion:
o A discussion of the other surrogate markers seems to be more applicable to the introduction (ie, people have tried these other things and were not good enough, so maybe our study will show an alternative that is better)
o The clinical application of these findings may be better mentioned earlier and made more succinct. It remains unclear how exactly this may be applied to clinical practice.
· Overall, this paper is a novel study that may provide an important piece of evidence in the treatment of male idiopathic infertility. At times, the text is nearly unintelligible due to poor grammar, unclear phrasing, and uncomfortable sentence structure. An English grammar check is highly recommended with a focus on the use of appropriate language and verb/subject agreement.
Author Response
Reviewer 2
In the manuscript titled Testosterone serum levels are related to sperm DNA fragmentation index reduction after FSH administration in males with idiopathic infertility, the authors performed a re-analysis of clinical trials data investigating the role of FSH supplementation in DNA fragmentation in men with idiopathic infertility. The manuscript may be useful for the advancement of the field; however, it needs some revisions prior to its acceptance for publication. The most relevant commentary relates to the methods section and requires a re-evaluation of the study design prior to publication. Detailed comments are as follows:
Introduction:
Section 1.1 – the mention of hypogonadotropic hypogonadism is out of place; how can this be tied into idiopathic male infertility in this section?
ANSWER: We mentioned the hypogonadotropic hypogonadism within the introduction section since the rationale treatment currently used in this clinical condition is considered the model for empirical therapies applied in male idiopathic infertility. We introduced in this section the following sentence, to link the two treatments: “Several empirical treatments have been suggested, trying to mirror the therapeutic approach applied to hypogonadotropic hypogonadism”
Moreover, we introduced the definition of male idiopathic infertility, focusing on the differences with unexplained infertility, as reported in the comment to the first reviewer.
Section 1.2 – this section contains a useful overview of prior literature but is a bit too long.
ANSWER: Thank you for your comment. We revised this paragraph, reducing the length.
Section 1.3 – what are the clinical implications of a high or low sDF index?
ANSWER: Sperm DNA fragmentation (sDF) index is a measure of the gamete DNA quality and integrity mainly, but not only, occurring during spermatogenesis, the different mechanisms have been identified as responsible for DNA fragmentation, including apoptosis, defect of chromatin remodelling, oxidative stress and many others (Sakkas 2009) . The higher the fragmentation, the lower is the competence of the sperm, i.e. the ability to fertilize the oocyte. Accordingly, there is wide literature suggesting that infertile men show higher sDF index compared to fertile ones. We comprehensively demonstrated this sDF diagnostic role in a meta-analysis (Santi D, Spaggiari G, Simoni M. Reprod Biomed Online. 2018 Sep;37(3):315-326. doi: 10.1016/j.rbmo.2018.06.023) detecting also an accurate threshold of 20% to discriminate fertile to infertile men. We included this aspect in the 1.3 section of the introduction.
Overall, the introduction is insufficient, mostly due to structure and wording. A better way to structure the introduction would be to describe the physiologic role of FSH, and the pathophysiology of DNA fragmentation and then bring it together as a “we don’t know how these two might be related”
ANSWER: Since other reviewers suggested to reduce introduction length, we considered to build the introduction from male idiopathic infertility definition, throughout the use of FSH in this condition, until the efficacy evaluated so far. In this way, without stretching the introduction, we could guide the reader to the aims of our analysis.
Methods:
o Please mention why these keywords were chosen and why these inclusion criteria were used.
ANSWER: We further explained the selection of keywords and inclusion criteria as follows: “Considering the rich literature on FSH application in male idiopathic infertility, key-words and inclusion criteria were set to detect only those studies in which both testosterone and sDF are reported after treatment”.
o The use of “systematic review” is confusing in this context. It is unclear if this systematic literature review was how the mechanism by which the clinical trials were selected. If so, clarify. If it is truly a systematic review, the studies included should be evaluated and an evaluation of these studies should be provided in the paper.
ANSWER: The term “systematic” identifies a specific type of methodological work, characterized by a rigorous and transparent approach for research studies, with the aim of making the literature search performed reproducible. Indeed, we reported the systematic structure of our review of the literature, in order to demonstrate that the methods used to search for and analyse the data are transparent, reproducible and defined before searching begins (Effectiveness of psychosocial interventions for reducing parental substance misuse. Ruth McGovern et al. https://doi.org/10.1002/14651858.CD012823.pub2). The systematic review of the literature is used in our work to be sure to detect all trials available in the literature to reach our aim.
o How does this study differ from a meta-analysis? If it is a meta-analysis, please rewrite it, but if it is a systematic review it needs to have the appropriate metrics included.
ANSWER: This work is not a meta-analysis, which, should refer to the methodological analysis of data extracted by a systematic review. We performed a systematic literature search in order to extract all studies useful to reach the final aim.
o Breaking this section into so many subsections is difficult to read. Unless this is required by the journal, please adjust to make larger, more cohesive sections.
ANSWER: thank you for this observation. We reduced the number of subheadings.
Results:
o Please include a PRISMA diagram
ANSWER: We included the PRISMA diagram as supplementary material.
o Please describe which analysis was used to determine the values reported in the first two paragraphs of section 3.1 (if it was all from the correlation analysis, can be mentioned once)
ANSWER: correlation analyses were performed by Spearman’s correlation, which reported the Rho, to measure the strength of the correlation, and the p-value to highlight its statistical significance. We added reference to this test in this section. All these results are obtained with the same statistical analysis.
Discussion:
o A discussion of the other surrogate markers seems to be more applicable to the introduction (ie, people have tried these other things and were not good enough, so maybe our study will show an alternative that is better)
ANSWER: Following also your previous comment, shifting these examples to the introduction section probably creates more confusion in the reader. In the discussion section we reported other surrogate markers of FSH efficacy since we detected a significant change in parameters not previously reported in the work. It is an unexpected result that requires discussion, looking for what is already published in the literature.
o The clinical application of these findings may be better mentioned earlier and made more succinct. It remains unclear how exactly this may be applied to clinical practice.
ANSWER: Thank you for your comment. Indeed, our results unravel new potential correlations and markers not previously evaluated sufficiently in this treatment management. Thus, it is difficult to define which could be its clinical application. We tried to introduce this aspect as follows: “This new correlation, although interesting, is far to be directly transposed to clinical practice. Specific properly designed prospective trials must be designed to understand the real clinical application of the finding.”
Overall, this paper is a novel study that may provide an important piece of evidence in the treatment of male idiopathic infertility. At times, the text is nearly unintelligible due to poor grammar, unclear phrasing, and uncomfortable sentence structure. An English grammar check is highly recommended with a focus on the use of appropriate language and verb/subject agreement.
ANSWER: We performed a English language revision.
Reviewer 3 Report
In this paper, Lispi and colleagues analyzed the correlation between FSH stimulation in idiopathic infertile men and sperm DNA fragmentation (sDF). They performed a retrospective post-hoc analysis on raw data of clinical trials, confirming the beneficial effect of FSH stimulation on spermatogenesis, highlighted by decease in sDF, of about 20%. The authors then hypothesized a communication/interaction between the two cell compartments of the testis (i.e. Sertoli and Leydig 48) in response to FSH administration.
The paper is interesting and well conducted, with some minor revisions to be addressed:
- I suggest the authors to consider their paper not as an article, but as a systematic review/meta-analysis, as they analyzed already-published papers;
- The abstract is too long and, following the journal instructions, it (as well as the introduction) should not contain subheadings;
- In the text, references should be put in square brackets, and not in round brackets.
Author Response
Reviewer 3
In this paper, Lispi and colleagues analyzed the correlation between FSH stimulation in idiopathic infertile men and sperm DNA fragmentation (sDF). They performed a retrospective post-hoc analysis on raw data of clinical trials, confirming the beneficial effect of FSH stimulation on spermatogenesis, highlighted by decease in sDF, of about 20%. The authors then hypothesized a communication/interaction between the two cell compartments of the testis (i.e. Sertoli and Leydig 48) in response to FSH administration.
The paper is interesting and well conducted, with some minor revisions to be addressed:
- I suggest the authors to consider their paper not as an article, but as a systematic review/meta-analysis, as they analyzed already-published papers;
ANSWER: We thak you for the suggestion but we respectfylly disagree with this comment. Indeed, our study is a systematic review but not a meta-analysis of previously published data. Indeed, we did not work on aggregate data, as occurs in meta-analyses, but we re-analysed raw data. This is a real re-analysis of data obtained in different clinical setting.
- The abstract is too long and, following the journal instructions, it (as well as the introduction) should not contain subheadings;
ANSWER: Thank you for your comment. We revised the abstract
- In the text, references should be put in square brackets, and not in round brackets.
ANSWER: Thank you for the comment. The manuscript has been properly formatted.